# Improving Antibody Design with Force-Guided Sampling in Diffusion Models

## Abstract

Antibodies, crucial for immune defense, primarily rely on complementarity-determining regions (CDRs) to bind and neutralize antigens, such as viruses. The design of these CDRs determines the antibody's affinity and specificity towards its target. Generative models, particularly denoising diffusion probabilistic models (DDPMs), have shown potential to advance the structure-based design of CDR regions. However, only a limited dataset of bound antibody-antigen structures is available, and generalization to out-of-distribution interfaces remains a challenge. Physics based force-fields, which approximate atomic interactions, offer a coarse but universal source of information to better mold designs to target interfaces. Integrating this foundational information into diffusion models is, therefore, highly desirable. Here, we propose a novel approach to enhance the sampling process of diffusion models by integrating force field energy-based feedback. Our model, DiffForce, employs forces to guide the diffusion sampling process, effectively blending the two distributions. Through extensive experiments, we demonstrate that our method guides the model to sample CDRs with lower energy, enhancing both the structure and sequence of the generated antibodies.

## 1 Introduction

Antibodies are key therapeutic proteins due to their ability to selectively bind to a variety of disease-causing antigens, including viruses. Antibodies consist of two heavy and two light chains, forming a Y-shaped structure. Critical to their ability to recognize diverse antigens are the six complementarity determining regions (CDRs) located at the tips of this structure. The diversity of antibodies is derived from the extensive combinatorial possibilities of these CDRs. A CDR of length $L$ can theoretically have up to $20^L$ different amino acid sequences, owing to the 20 types of amino acids that can be placed at each position. Therefore, a key step in developing therapeutic antibodies is designing effective CDRs that specifically bind to target antigens (Kunik et al., 2012; Sela-Culang et al., 2013).

Traditional approaches to antibody design predominantly rely on animal immunization and computational methods. Animal immunization is inherently limited to the production of naturally occurring antibodies and raises ethical concerns (Gray et al., 2020), despite its effectiveness in generating high-affinity antibodies. Traditional *in silico* methods, on the other hand, utilize complex biophysical energy functions (Warszawski et al., 2020; Adolf-Bryfogle et al., 2018) to predict how potential antibodies might interact with their targets. However, they depend on expensive simulations, are prone to convergence to local optima, and possess inherent limitations due to the complex nature of interactions which cannot be efficiently represented by basic statistical functions (Graves et al., 2020). This situation underscores the need for alternative approaches in antibody design.

Recently, denoising diffusion probabilistic models (DDPMs) have emerged as a powerful technique for learning and sampling from complex, high-dimensional protein distributions (Watson et al., 2023; Yim et al., 2023; Trippe et al., 2023). In particular, this advancement has shown potential in the structure-based design of CDRs. Recent work (Luo et al., 2022; Martinkus et al., 2023) has demonstrated the capabilities of diffusion models for modeling the CDRs of antibodies at the atomic level, conditioned on the antigen and an antibody framework. However, the available dataset of bound antibody-antigen structures is limited, and generalization to out-of-distribution interfaces remains a challenge. While diffusion models provide accurate approximations within the known distribution, they struggle with out-of-distribution scenarios. This limitation poses a challenge for advancing CDR

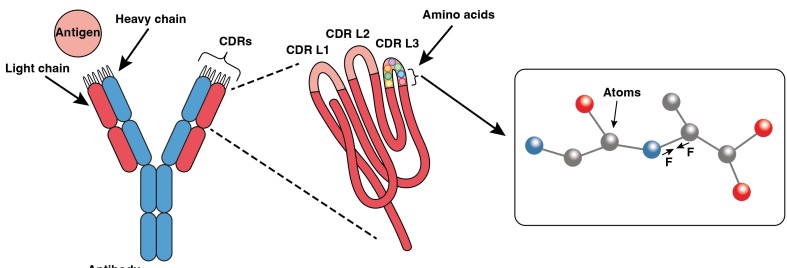

Figure 1: The antigen-binding region comprises six complementarity-determining regions (CDRs). Each CDR is constructed from a variety of amino acids, which are themselves made up of atoms. These atoms are governed by forces, denoted by the symbol $F$.

design as many antibodies generated *in silico* with diffusion models fail to demonstrate functionality *in vitro* (Shanehsazzadeh, 2024; Zeni et al., 2023; Sidhu & Fellouse, 2006).

To address this challenge, we propose DIFFFORCE, a force-guided DDPM sampling method inspired by traditional physics-based simulation techniques such as molecular dynamics (MD). Physics-based force fields, which approximate atomic interactions (as shown in Figure 1), provide a coarse but universal source of information to better align antibody designs with target interfaces. Integrating this foundational data into diffusion models overcomes the limitations of distribution learning, as physics-based models generalize well despite being poor approximators. By combining these approaches, we enhance the ability to model out-of-distribution interfaces as we are guided by force field energy, while the structural *antibody-like* details are left to be determined by the diffusion model. While previous studies have used force field-based functions to refine antibody structures after diffusion generation (Luo et al., 2022), or have trained separate networks to approximate the forces for guiding an unconditional diffusion model (Wang et al., 2024), we are the first to construct a principled method of force-guided DDPM sampling, effectively blending the two distributions. Given a protein complex consisting of an antigen and an antibody framework as input, we first initialize the CDR with arbitrary positions, sequence and orientations. Then, during the sampling stage, we iteratively update the atom positions guided by the gradients of force field energy, which are calculated for the denoised sample approximation. We highlight our main contributions as follows:

- We introduce the first force-guided diffusion model, which utilizes a differentiable force field to guide the sampling process, effectively leveraging the weighted geometric mean of the two distributions. Unlike existing methods, our model does not require to train a separate network for energy approximation or condition the diffusion model on energy.

- We propose a method to approximate the denoised sample of antibody atom coordinates, offering an elegant interpolative interpretation. This enables accurate energy computation, ensuring the precise application of forces during diffusion sampling. We also present an approach for approximating the denoised samples of amino acid types and orientations.

We evaluate our model on the CDR sequence-structure co-design task. We show that our proposed method effectively guides the model to sample CDRs with lower energy, outperforming several state-of-the-art models. We observe that our model generates more favorable structures earlier in the sampling process, leading to an enhanced quality of produced antibody sequences.

## 2 RELATED WORK

**Diffusion Models for Antibody Design**   Antibody design involves creating the sequence and structure of antibodies that can bind to target antigens. This process differs from general protein design, where sequences are derived from known structures (Dauparas et al., 2022; Ingraham et al., 2019), or structures are predicted based on amino acid sequences (Jumper et al., 2021). In antibody design, the sequences and structures of the CDRs are usually initially unknown. While various generative models have been proposed to learn such data distribution, diffusion models (Sohl-Dickstein et al., 2015; Dhariwal & Nichol, 2021) have recently gained prominence for their

effectiveness in ensuring stable training and achieving good distribution coverage. Diffusion models achieve state-of-the-art performance in antibody design by learning to generate new data through denoising samples from a prior distribution. The DiGress model (Vignac et al., 2023) demonstrated how to utilize a discrete diffusion process for molecules, while the work of DiffAb (Luo et al., 2022) proposed the first diffusion model to perform joint design of sequence and structure of the antibody CDR regions while conditioning on the antigen-antibody complex. AbDiffuser (Martinkus et al., 2023) improved this further by incorporating strong priors and being more memory efficient with side chain generation. However, these models still face challenges in accurately modeling the complex interactions within antigen-antibody interfaces, particularly when dealing with out-of-distribution data.

**Guided Generation** Guiding generative models to produce specific outcomes is highly desirable for a variety of applications (Ho et al., 2022; Nichol et al., 2023). To achieve this, two main methods have been proposed, a classifier guidance (Dhariwal & Nichol, 2021; Song et al., 2021b) and a classifier-free guidance (Ho & Salimans, 2021). Recently, a concurrent work (Wang et al., 2024) introduced a force-guided diffusion model to produce protein conformations aligned with Boltzmann's equilibrium distribution, based on the classifier guidance approach. However, this method requires training an additional network to approximate the intermediate force vector to guide an unconditional model, which can result in inaccurate estimates. In contrast, our method employs a differentiable force field for guided sampling, eliminating the need for a separate energy approximation network and ensuring more accurate energy calculations. Additionally, a loss guidance approach has been proposed (Song et al., 2023), leveraging differentiable loss functions to guide the model without additional training on noisy paired data. Similarly, our approach uses a differentiable force field to guide the sampling.

## 3 METHOD

We propose force-guided DIFFFORCE, a diffusion model targeting CDR region generation for antibodies. Building upon the DIFFAB diffusion model introduced in Section 3.1, we present a novel strategy in Section 3.2 that integrates force guidance into the diffusion model's sampling. By employing force to guide the sampling process, DIFFFORCE achieves CDRs with lower energy, leading to an improved structure and ultimately the sequence of the generated antibodies. A visualization of the method is shown in Figure 2.

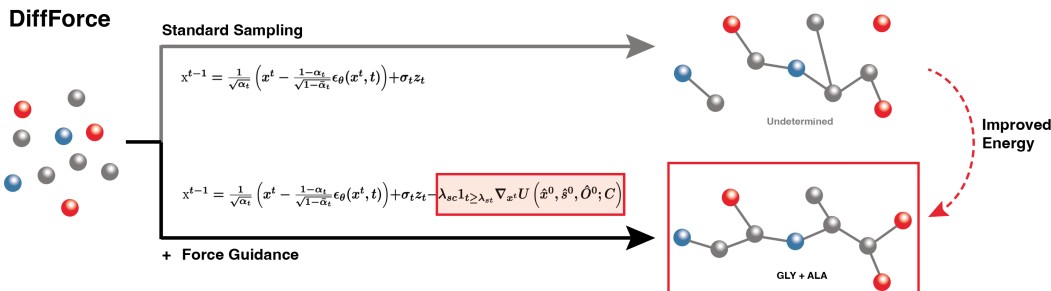

Figure 2: Antibody CDR generation with different sampling strategies. **Upper**: Standard DDPM sampling without force guidance. **Lower**: Incorporating force guidance into sampling, the model generates CDR structures with lower energy. Notation explained in the main text.

### 3.1 DIFFUSION MODEL

Our model builds upon the DIFFAB diffusion model (Luo et al., 2022). DIFFAB represents each amino acid in an antibody by its type $s_i \in \{A \dots Y\}$, the coordinates of its $C_\alpha$ atom $x_i \in \mathbb{R}^3$, and its orientation $O_i \in SO(3)$. Assuming that the structures of the antigen, the antibody framework, and five other CDRs are known, it designs one CDR loop at a time, denoted as $R = \{(s_j, x_j, O_j) \mid j = l+1, \dots, l+m\}$, given the rest of the antibody-antigen complex $C = \{(s_i, x_i, O_i) \mid i \neq j\}$, which includes a set of five fixed CDRs.

The forward diffusion process from $t = 0$ to $T$, is Markovian and incrementally adds noise to three different modalities using non-learnable distributions $q$: The $C_\alpha$ atom positions follow a Gaussian distribution, $q(x_j^t \mid x_j^0)$; amino acid types follow a multinomial distribution, $q(s_j^t \mid s_j^0)$; and the orientations of amino acids follow an isotropic Gaussian distribution, $q(O_j^t \mid O_j^0)$. The backward diffusion process (from $t = T$ to 0), refines each modality back towards the original data distribution. The reverse process is guided by learnable models $p_\theta$, which approximate the posterior distributions at each step using three distinct neural networks (further denoted as $F, G, H$, respectively) for the three modalities. For more details on the DIFFAB model, see Section 3 of the original paper (Luo et al., 2022), and for additional information on DDPMs, refer to Appendix A.

## 3.2 FORCE GUIDED ANTIBODY DESIGN

### 3.2.1 FORCE FIELD

Molecular dynamics (MD) simulations provide insights into the dynamic behavior of molecular systems by numerically integrating Newton's equations of motion (Chandler et al., 1987) for $N$ particles:

$$m_i \frac{d^2 x_i}{dt^2} = F_i = -\frac{\partial}{\partial x_i} U(x_1, x_2, \ldots, x_N), \tag{1}$$

where $m_i$, $x_i$, and $F_i$ represent the mass, position, and force on each particle, respectively. The energy $U(x_1, x_2, \ldots, x_N)$ is a function of the coordinates of all $N$ particles. By solving Newton's equation, MD simulations approximate the evolution of molecular systems over time.

An MD force field is a parametrised function used to evaluate the energy $U(x_1, x_2, \ldots, x_N)$ of a given configuration. For proteins, these forcefields are typically empirical, due to the large system sizes, and their functional forms and parameters are tuned to closely match experimental observations. Common terms include both bonded interactions, such as bond stretching, angle bending, and torsional angles, and non-bonded interactions, like van der Waals forces and electrostatic interactions.

In the context of antibody design, the force field takes a protein $P$ (e.g., set of atom coordinates $x$) and computes the energy $U$. By calculating the gradient $\nabla U$, we can determine how $U$ varies with changes in atomic positions. This gradient indicates how to adjust each atom's position to minimize the total energy of the protein structure. Lower energy configurations often correspond to more thermodynamically stable antigen-antibody complexes, which are associated with higher affinity (Ji et al., 2023). Using the relationship between energy and force, $-\nabla U(x) = F$, we can simulate the equations of motion to evolve this dynamical system according to the energy $U$.

### 3.2.2 DIFFFORCE-$C_\alpha$ : INTERPOLATING BETWEEN $p_{\text{data}}$ AND $e^{-\kappa U(x_0; C)}$

For simplicity, we consider the setting where the residues are fixed, and our focus is to guide the $C_\alpha$ atom coordinates with a prescribed force field. Rather than sample unconditionally from the data distribution, we are interested in sampling from the following tilted distribution:

$$\pi_0(x_0) = \frac{p_{\text{data}}(x_0) e^{-\kappa U(x_0; C)}}{\int p_{\text{data}}(x_0) e^{-\kappa U(x_0; C)} \mathrm{d}x_0}, \tag{2}$$

where we use the notation $U(x_0; C)$ to denote that $C$ is fixed throughout simulation. This induces a new distribution we wish to sample from that interpolates between the Boltzmann distribution $e^{-\kappa U(x_0)}$ [1] and $p_{\text{data}}(x_0)$. One way to interpret this is to think of $p_{\text{data}}(x_0)$ as a prior and $e^{-\kappa U(x_0)}$ acting as a likelihood of the form $p(y|x_0)$. Thus $\pi_0(x_0)$ is akin to a posterior of the form $p(x_0|y)$ that is in a way conditioned to make the binding energy small. However, unlike (Song et al., 2023; Komorowska et al., 2024), we do not have an explicit notion of the variable $y$ in this setting. We highlight that (Wang et al., 2024) concurrently explore an akin setting; however, their approach is focused on learning a new modified score while our is focused on approximations during inference.

An alternate and akin approach is to construct $\pi_0$ as a log-concave interpolation, as in annealed sampling (Neal, 2001), that is to form the weighted geometric mean $\pi_0 \propto p_{\text{data}}^{1-\beta} \exp(-\kappa U(x_0))^\beta$ for $\beta \in [0, 1]$. This has the interpretation that we are now trying to sample from a distribution that is an

---

[1]For brevity we have dropped the conditioning on $C$.

interpolation between $p_{\text{data}}(x_0)$ and $e^{-\kappa U(x_0)}$. By leveraging the weighted geometric mean of the distributions, we ensure that if one distribution suggests a particular outcome is extremely unlikely, it influences the other, thus pulling the combined distribution towards more realistic outcomes. This method aligns well with our goal of generating high-quality samples with good binding energies, providing a balanced compromise between the two. In practice, however, we follow Equation 2 as it provides a form that is easier to tune and more in line with prior works on conditioning diffusion models. Due to this connection, we will refer to $\pi_0$ as the interpolating distribution. To sample from Equation 2 we estimate the interpolating score $\nabla_{x_t} \ln \pi_t(x_t)$ (Chung et al., 2023):

$$\nabla_{x_t} \ln \pi_t(x_t) = \nabla \ln \int \pi_0(x_0)p(x_t|x_0)dx_0, \tag{3}$$

$$= \nabla_{x_t} \ln \int e^{-\kappa U(x_0;C)}p_{\text{data}}(x_0)p(x_t|x_0)dx_0, \tag{4}$$

$$= \nabla_{x_t} \ln \int e^{-\kappa U(x_0;C)}p(x_0|x_t)dx_0 + \nabla_{x_t} \ln p(x_t), \tag{5}$$

where $p(x_0|x_t)$ is the transition density of the backwards SDE (the denoising process), which we do not have access to. Following (Komorowska et al., 2024; Chung et al., 2023), we approximate it with a point mass centered at its mean:

$$\int e^{-\kappa U(x_0;C)}p(x_0|x_t)dx_0 \approx \int e^{-\kappa U(x_0;C)}\delta_{\mathbb{E}[x_0|x_t]}(x_0)dx_0 \tag{6}$$

$$= e^{-\kappa U(\mathbb{E}[x_0|x_t];C)}. \tag{7}$$

Then, the approximate interpolating score is given by $\nabla_{x_t} \ln \pi_t(x_t) \approx -\kappa\nabla_{x_t}U(\mathbb{E}[x_0|x_t]) + \nabla_{x_t} \ln p(x_t)$, and we can use Tweedie's formula (Robbins, 1992) to compute $\mathbb{E}[x_0|x_t]$ given we have a good approximation of the score:

$$\mathbb{E}[x_0 \mid x_t] = \frac{x_t + (1 - \bar{\alpha}_t)\nabla_{x_t} \ln p_t(x_t)}{\sqrt{\bar{\alpha}_t}} \approx \hat{x}_0(x_t) = \frac{1}{\sqrt{\bar{\alpha}_t}}\left(x_t - \sqrt{1 - \bar{\alpha}_t}\epsilon_\theta(x_t, t)\right). \tag{8}$$

Here, $\bar{\alpha}_t = \prod_{\tau=1}^{t} \alpha_\tau = \prod_{\tau=1}^{t}(1 - \beta_\tau)$, where $\beta_t$ is the cosine variance schedule for the diffusion model, and $\epsilon_\theta$ is the standard Gaussian noise added to the $x_t$ predicted by the neural network $F$. This yields the following sampler, with $z_t$ denoting standard Gaussian:

$$x_{t-1} = \frac{1}{\sqrt{\alpha_t}}\left(x_t - \frac{1 - \alpha_t}{\sqrt{1 - \bar{\alpha}_t}}\epsilon_\theta(x_t, t)\right) + \sigma_t z_t - \kappa\nabla_{x_t}U(\hat{x}_0(x_t)). \tag{9}$$

We now have ingredients to generate an approximate sample from the interpolating distribution $\pi_0$.

### 3.2.3 FORCE GUIDANCE FOR RESIDUE TYPES

We have derived an approach to approximate the $C_\alpha$ atom coordinates at $t = 0$, further denoted as $\hat{x}^0$. However, we also need to devise approximations for the amino acid types and orientations to obtain an estimate for $\mathbb{E}[R^0|R^t]$, which is required to calculate the energy $U$. Unlike the $C_\alpha$ coordinates, the approximations for amino acid types and orientations do not follow Tweedie's formula. To account for it, we derive an alternative approach to estimate $\hat{s}^0$ and $\hat{O}^0$ using the settings provided.

**Amino Acid Types** The generative diffusion process for amino acid types, denoted by $p(s_j^{t-1}|R^t, C)$ and defined in (Luo et al., 2022, Equation 3), is designed to approximate the posterior $q(s_j^{t-1}|s_j^t, s_j^0)$. This alignment is quantified using the Kullback–Leibler (KL) divergence, as suggested in (Hoogeboom et al., 2021, Equation 15):

$$\text{KL}(q(s^{t-1}|s^t, s^0)\|p(s^{t-1}|s^t)) = \text{KL}\left(C(\theta_{\text{post}}(s^t, s^0))\|C(\theta_{\text{post}}(s^t, \hat{s}^0))\right), \tag{10}$$

where the KL divergence is minimized when the parameterized posterior $\theta_{\text{post}}(s^t, s^0)$ is equivalent to $\theta_{\text{post}}(s^t, \hat{s}^0)$ thus making $\hat{s}^0$ a good predictor for $s^0$ given we observe $s^t$. Following this, we can derive the distribution for the posterior sample at timestep $t - 1$ as:

$$q(s_j^{t-1}|s_j^t, s_j^0) = \text{Multinomial}\left(\left[\alpha_{\text{type}}^t \cdot \text{onehot}(s_j^t) + (1 - \alpha_{\text{type}}^t) \cdot \frac{1}{20}\right]\right.$$
$$\left.\odot\left[\bar{\alpha}_{\text{type}}^{t-1} \cdot \text{onehot}(s_j^0) + (1 - \bar{\alpha}_{\text{type}}^{t-1}) \cdot \frac{1}{20}\right]\right). \tag{11}$$

Here $\bar{\alpha}_{\text{type}}^t = \prod_{\tau=1}^t (1 - \beta_{\text{type}}^\tau)$ and $\beta_{\text{type}}^t$ is the probability of uniformly resampling another amino acid from among the 20 types. The neural network $G$ is tasked with predicting $s_j^0$, leveraging the learned distributional characteristics of amino acid types. In order to approximate the denoised sample for amino acid types at $t = 0$, namely $\hat{s}^0$, the idea is to utilize only the second term of Equation 11:

$$\hat{s}_j^0 = \bar{\alpha}_{\text{type}}^{t-1} \cdot \text{onehot}(s_j^0) + (1 - \bar{\alpha}_{\text{type}}^{t-1}) \cdot \frac{1}{20}, \tag{12}$$

where $\hat{s}_j^0$ predicts the amino acid type at $t = 0$ for each amino acid $j$.

**Amino Acid Orientations**   The denoising process for amino acid orientations is captured via SO(3) elements, as described by (Leach et al., 2022) and implemented by (Luo et al., 2022, Equation 11):

$$p(O_j^{t-1}|R^t, C) = \mathcal{IG}_{\text{SO(3)}}\left(O_j^{t-1}|H(R^t, C)[j], \beta_{\text{ori}}^t\right), \tag{13}$$

where $H$ is a neural network that denoises the orientation matrix for amino acid $j$, $\mathcal{IG}_{\text{SO(3)}}$ denotes the isotropic Gaussian distribution on SO(3) parameterized by a mean rotation and a scalar variance, $\beta_\tau^t$ is the variance increase with the step $t$. To obtain the approximation $\hat{O}_j^0$ for amino acid orientation, we propose an approach of iteratively denoising the sample $O_j^t$ for $t$ iterations, where each iteration predicts the sample $O_j^{t-1}$. Namely, by iteratively applying Equation 13 until timestep $t$ reaches 0 for each amino acid $j$, we converge to the approximation $\hat{O}_j^0$, effectively reversing the forward diffusion:

$$\hat{O}_j^0(R^t) \approx \tilde{O}_j^0 \sim p(O_j^{t-1}|R^t, C) \prod_{s=1}^{t-1} p(O_j^{s-1}|R^s, C). \tag{14}$$

We have now obtained denoised approximations of atom coordinates, amino acid types, and orientations. This estimate can be utilized in further algorithms to compute the energy $U$.

### 3.2.4   IMPLEMENTATION

We derive a novel approach for guiding the sampling of $C_\alpha$ atom coordinates with a prescribed force:

---
**Algorithm 1** DIFFFORCE-$C_\alpha$ Sampling with Force Guidance

---
1: $x^T \sim \mathcal{N}(0, I)$
2: **for** $t = T, \ldots, 1$ **do**
3:     $z \sim \mathcal{N}(0, I)$ if $t > 1$, else $z = 0$
4:     estimate $\hat{x}^0(R^t)$ using Eq 8, $\hat{s}^0(R^t)$ using Eq 12 and $\hat{O}^0(R^t)$ using Eq 14
5:     $x^{t-1} = \frac{1}{\sqrt{\alpha_t}} \left( x^t - \frac{1-\alpha_t}{\sqrt{1-\bar{\alpha}_t}} \epsilon_\theta(x^t, t) \right) + \sigma_t z_t - \lambda_{sc} \mathbb{1}_{t \geq \lambda_{st}} \nabla_{x^t} U\left( \hat{x}^0, \hat{s}^0, \hat{O}^0; C \right)$
6:     $R^{t-1} = \left( x^{t-1}, s^{t-1}, O^{t-1} \right)$, sample $s^{t-1}, O^{t-1}$ following (Luo et al., 2022)
7: **end for**
8: **return** $x^0, s^0, O^0$

---

We introduce two hyperparameters, force scale ($\lambda_{sc}$) and force start ($\lambda_{st}$). The $\lambda_{sc}$ parameter dictates the magnitude of the force, gradually adjusted from 0.0 to $\lambda_{sc}$ using a linear scheduling strategy. This parameter is applied to normalized forces as detailed in Appendix B. The $\lambda_{st}$ parameter defines when force application begins, with a value of 0.3 indicating initiation of force at 70% of the sampling.

## 4   EXPERIMENTS

We evaluate the effectiveness of the DIFFFORCE model on the CDR sequence-structure co-design task. We demonstrate that 1) force guidance effectively guides the model to sample CDRs with lower energy; 2) using force guidance, DIFFFORCE outperforms current state-of-the-art models by generating high-quality antibody samples, with an emphasis on the CDR H3 region.

### 4.1   EXPERIMENTAL SETUP

**Baselines**   We compare DIFFFORCE against two baseline models, the diffusion model DIFFAB (Luo et al., 2022) and the traditional energy-based method RABD (Adolf-Bryfogle et al., 2018). Both baseline models are evaluated using default settings. For more details, see Appendix F.

**Dataset and Diffusion Model** To evaluate our model, we use the SAbDab database (Dunbar et al., 2013) (with Chothia numbering scheme), filtering out complexes with resolutions worse than 4Å and those targeting non-protein antigens. Following (Luo et al., 2022), we cluster antibodies based on 50% H3 sequence identity and select five clusters for the test set, comprising 19 complexes. We use the *codesign_single* pre-trained model from DIFFAB, which generates one CDR region at a time.

**Energy** Energy of protein structures is evaluated using MadraX (Orlando et al., 2023), which provides the Gibbs free energy ($\Delta G$) of the complex. Unlike other force fields, such as Rosetta (Alford et al., 2017) or FoldX (Schymkowitz et al., 2005), Madrax is fully differentiable. MadraX evaluates several categories of interaction energies, adapting 7 categories from FoldX (Schymkowitz et al., 2005) into a differentiable format. The energy considers the full protein structure, whose reconstruction is described in Appendix D. The energy is reported in *kcal/mol*.

**Metrics** To validate our model's performance, we use three key metrics; *1) Binding Energy Improvement (IMP)* is calculated as the percentage of designed CDRs that show a reduction (improvement) in free binding energy ($\Delta\Delta G$) compared to the reference CDRs, indicating a stronger interaction with the target antigen. This evaluation uses the InterfaceAnalyzer from Rosetta (Alford et al., 2017). *2) Root Mean Square Deviation (RMSD)* measures the average spatial discrepancy between the $C_\alpha$ atoms of the generated and reference antibody structures, with a higher RMSD indicating greater structural diversity. *3) Amino Acid Recovery Rate (AAR)* is defined as the overlapping ratio of the generated sequence to the ground truth, evaluating how accurately the generated CDR sequences replicate the reference sequences (Adolf-Bryfogle et al., 2018).

## 4.2 RESULTS

We evaluate the performance of DIFFFORCE model on the sequence-structure co-design as introduced by (Luo et al., 2022), where the reference CDR is removed from the antibody-antigen complex. The diffusion model is therefore conditioned on antibody framework and antigen. For each antigen-antibody complex, we generate $n = 25$ samples for 3 heavy chain CDRs (HCDRs). We choose to focus on the heavy chain since it typically exhibits greater variability and influences on binding affinity compared to the light chain (López-Requena et al., 2007). The samples are produced through 100 generative timesteps ($T = 100$), with each sample maintaining the same length as its corresponding reference CDR in the test set. Finally, the generated structures, as well as reference original ones, are relaxed using OpenMM (Eastman et al., 2017) and Rosetta (Alford et al., 2017).

Table 1 shows that DIFFFORCE model recovers all three HCDRs sequences with greater accuracy (higher AAR) than both DIFFAB and RABD. Furthermore, DIFFFORCE achieves improved binding scores (higher IMP) for the H1 and H3 regions. The model exhibits RMSDs comparable to those of DIFFAB. Overall, the most substantial improvement is observed in the H3 region, which can be attributed to its significantly longer sequence and the smaller variability present in the H1 and H2 regions. This length allows for a wider range of adjustments and provides a greater scope for applying force guidance during sampling. This test validates the efficacy of our model in generating high-quality CDRs, with an emphasis on handling the complex CDR H3 region.

| Method | AAR (%) ↑ | | | IMP (%) ↑ | | | RMSD (Å) ↓ | | |
|---|---|---|---|---|---|---|---|---|---|
| | H1 | H2 | H3 | H1 | H2 | H3 | H1 | H2 | H3 |
| RABD | 22.85 | 25.50 | 22.14 | 43.88 | **53.50** | 23.25 | 2.261 | 1.641 | **2.900** |
| DIFFAB | 58.70 | 49.37 | 26.08 | 47.91 | 30.77 | 23.59 | **1.438** | **1.235** | 3.605 |
| DIFFFORCE | **60.78** | **53.51** | **29.52** | **49.45** | 36.81 | **30.22** | 1.561 | 1.401 | 3.612 |

Table 1: Results on CDR sampling. The best result for each metric is highlighted in **bold**.

Figure 3 presents three generated CDR samples using DIFFFORCE. The binding specificity is determined by the interaction between the antibody's paratope region and the antigen's epitope region (Peng et al., 2014). The paratope region, comprising the interacting amino acid residues from a specific CDR region of an antibody, is highlighted in blue. The epitope, defined as the antigen residues within $<= 5$Å of the CDR, is marked in red. The antibodies target the SARS-CoV-2 RBD antigen, potentially offering a treatment strategy for COVID-19 (Law et al., 2021).

We focus on visualizing the CDR-H3 region, as it is often the most variable part of the antibody, determining its precise binding capability to a wide range of antigens and playing a key role in the immune response to pathogens (Regep et al., 2017). The antigen-antibody framework is obtained from PDB:7DK2. All three samples show enhanced binding energy ($\Delta\Delta G$) as measured by Rosetta, despite significant structural deviations from the reference. This implies that a larger RMSD in the predicted CDR structure might indicate a viable alternative with enhanced binding capabilities, rather than a flaw in the prediction. Notably, Sample 1, which exhibits the best binding energy, appears to conform the best to the antigen, underscoring the potential advantages of structural deviations.

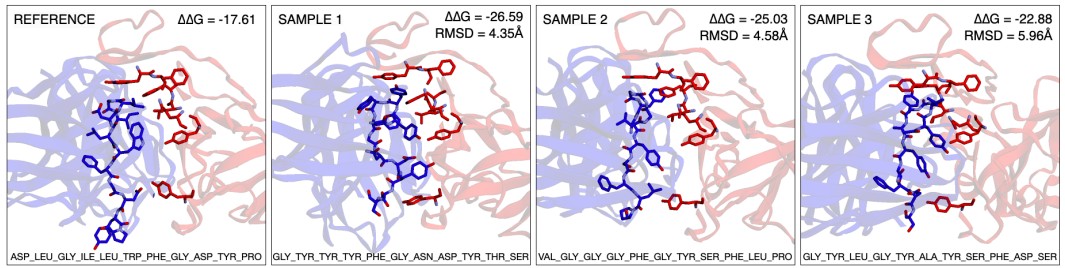

Figure 3: Generated samples for the CDR-H3 region of the PDB:7DK2 antigen-antibody complex. The RMSD, binding energy ($\Delta\Delta G$), and amino acid sequences are reported. The antigen is in red, and the antibody in blue. All samples show improved binding over the reference structure.

## 4.3 ANALYSIS

We conduct experiments to evaluate DIFFFORCE's performance in generating antibodies, focusing on energy and structure. Our findings highlight two key insights: 1) DIFFFORCE consistently demonstrates improved stability over DIFFAB, indicated by lower energy (Section 4.3.1); 2) DIFFFORCE achieves better structural conformity earlier in the sampling than DIFFAB (Section 4.3.2).

### 4.3.1 ENERGY LANDSCAPE

In a proof-of-concept study, we demonstrate that the proposed force guidance enhances the efficacy of the DIFFFORCE model. This approach generates CDR conformations with lower energy, indicating increased structural stability compared to DIFFAB. Specifically, we analyze the 7DK2 antigen-antibody complex, focusing on the heavy chain CDR regions H1, H2, and H3, using hyperparameters $\lambda_{sc} = 0.05$ and $\lambda_{st} = 0.3$. We compare the results of $n = 25$ samples, all starting from the same configuration at timestep 70 of the 100-timestep sampling process. The data is smoothed using a 10-period moving average, and energy is measured using MadraX. As shown in Figure 4, the results indicate a decrease in energy for both models, with DIFFFORCE consistently exhibiting lower energy values from $t = 30$ onward. This suggests that the force guidance in DIFFFORCE effectively directs the sampling, leading to more energetically favorable conformations and more stable antigen-antibody interactions. For details on hyperparameter choices and for other complexes, refer to Appendix G.

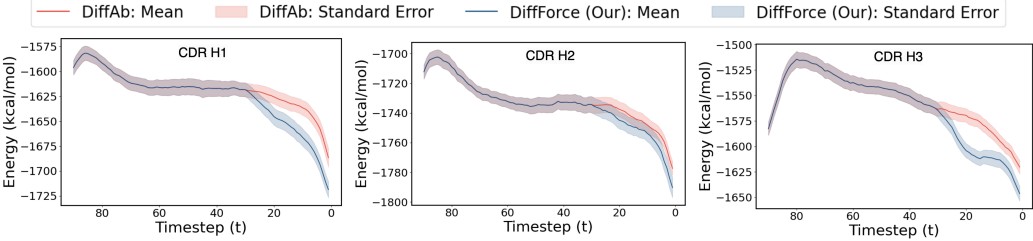

Figure 4: Energy of the PDB:7DK2 antigen-antibody complex's HCDR regions. Mean and standard error are based on $n = 25$ samples. The DIFFFORCE converges to lower energy levels than DIFFAB.

### 4.3.2 STRUCTURAL CONFORMITY

To further validate the effectiveness of force guidance, we conduct experiments on the structural conformity of generated antibody samples using the DIFFFORCE and DIFFAB models, focusing on the CDR H3 region of the 7DK2 antigen-antibody complex. We set hyperparameters at $\lambda_{sc} = 0.1$ and $\lambda_{st} = 0.3$, maintaining consistent seed values across both models. Figure 5 compares the models' performance at various sampling stages. Early in the diffusion process, DIFFFORCE consistently produces structures with better atomic coherence, fewer steric clashes, and higher structural connectivity than DIFFAB, particularly noticeable at earlier timesteps (e.g., $t = 15$, $t = 10$), indicating better sample fidelity. Additionally, DIFFFORCE achieves better energy at all sampled timesteps, demonstrating faster convergence to energetically optimal configurations. These empirical results highlight the potential of force guidance in improving the structural outcomes of diffusion-based antibody design, as well as reducing the need for post-generation relaxation.

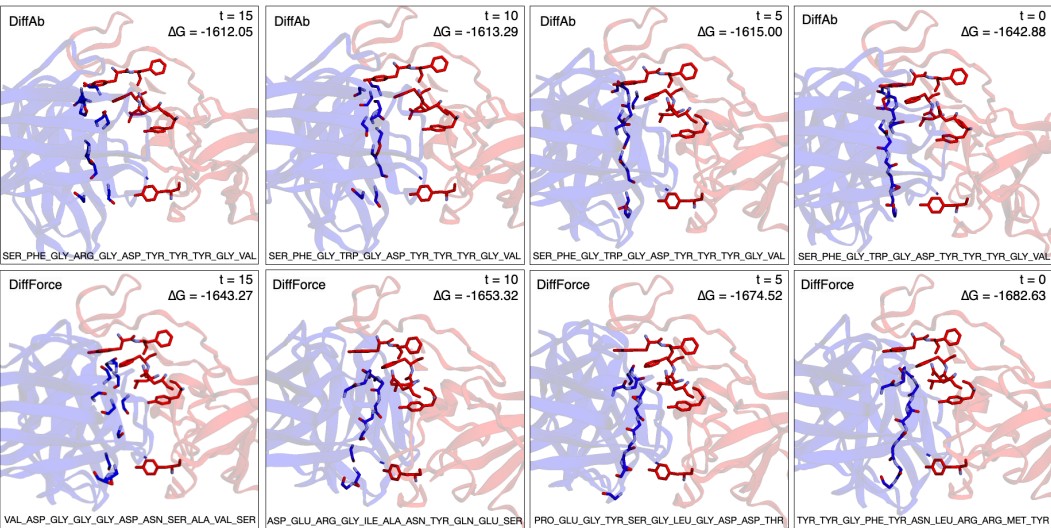

Figure 5: Results for the PDB:7DK2 complex's CDR-H3 region. Samples for DIFFAB (top) and DIFFFORCE (bottom) at timesteps $t = [15, 10, 5, 0]$. The energy and amino acid sequence are reported. DIFFFORCE achieves better structure and lower energy earlier in the sampling.

## 5 CONCLUSIONS AND FUTURE WORK

Antibodies play a vital role in the immune system by identifying and neutralizing antigens, such as viruses. Inspired by the fact that integrating physics-based force fields with generative models can improve out-of-distribution generalization for antibody design, we introduce DIFFFORCE, a diffusion model that incorporates force guidance into the sampling. Unlike existing methods, our model does not require conditioning a diffusion model on energy or training a separate network to approximate energy. We demonstrate that our model effectively guides the diffusion sampler to generate CDRs of better energy, outperforming several state-of-the-art models. This results in improved structure earlier in the sampling and enhances the sequences of the generated antibody CDRs.

While DIFFFORCE demonstrates promising results, it focuses on CDR sequence-structure co-design, with future potential in designing antibodies without bound framework structures. Moreover, the generated samples will require wet-lab experiments to confirm efficacy. Despite these and other limitations discussed in Appendix H, our work represents the first attempt to directly integrate a differentiable force field within diffusion sampling, effectively blending two distributions together.

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

## A  Denoising Diffusion Probabilistic Models

Denoising diffusion probabilistic models (DDPMs), introduced by Ho et al. (2020), represent a class of generative models that generate data by reversing a diffusion process. This process involves gradually transforming a sample from a simple distribution, like Gaussian noise, into a complex data distribution through learned reverse diffusion steps. The forward process incrementally adds noise to the data over a series of steps, transforming an initial data distribution into a distribution that is approximately Gaussian. This process is designed as a Markov chain, where each state $x_t$ only depends on the immediate previous state $x_{t-1}$. The transition from $x_{t-1}$ to $x_t$ is defined as:

$$x_t = \sqrt{\alpha_t} x_{t-1} + \sqrt{1 - \alpha_t}\epsilon, \quad \epsilon \sim \mathcal{N}(0, I). \tag{15}$$

In this equation, $\alpha_t$ (where $0 < \alpha_t \leq 1$) is a predefined variance schedule decreasing over time, which determines the proportion of the original data and noise at each step. The variable $\epsilon$ represents isotropic Gaussian noise, introducing randomness into the process. The reverse process aims to reconstruct the original data by sequentially removing the noise added during the forward process. This is achieved by training a neural network to estimate the original data distribution at each previous timestep, effectively learning the reverse of the forward process. The transition from noisy data $x_t$ back to less noisy data $x_{t-1}$ is modeled as:

$$p(x_{t-1}|x_t) = \mathcal{N}(x_{t-1}; \mu_\theta(x_t, t), \Sigma_\theta(x_t, t)). \tag{16}$$

Here, $\mu_\theta(x_t, t)$ and $\Sigma_\theta(x_t, t)$ are the mean and covariance of the Gaussian distribution for $x_{t-1}$, parameterized by a neural network with parameters $\theta$. These parameters are learned during training to minimize the difference between the actual noise and the predicted noise. The training of a DDPM is based on optimizing the variational lower bound, which effectively focuses on predicting the noise $\epsilon$ added at each step of the forward process. The loss function is defined as:

$$\mathcal{L}(\theta) = \mathbb{E}_{t,x_0,\epsilon} \left[ \|\epsilon - \epsilon_\theta(x_t, t)\|^2 \right]. \tag{17}$$

This loss function measures the mean squared error between the actual noise $\epsilon$ and the noise estimated by the neural network $\epsilon_\theta$. Successful training minimizes this error, enhancing the model's ability to reverse the diffusion process and, thereby, accurately generate samples that resemble the training data.

Song et al. (2021a) state that DDPMs is an example from the larger class of score-based models. They demonstrated that the discrete forward and reverse diffusion processes have their continuous time equivalents, that is, forward Stochastic Differential Equation, namely:

$$dx = -\frac{1}{2}\beta(t)x_t dt + \sqrt{\beta(t)}dw, \tag{18}$$

and it's reverse:

$$dx_t = \left[ -\frac{1}{2}\beta(t)x_t - \beta(t)\nabla_x \ln p_t(x_t) \right] dt + \sqrt{\beta(t)}d\bar{w}_t, \tag{19}$$

where the quantity $\nabla_{x_t} \ln p_t(x_t)$ is called the score and is closely related to the noise in DDPM by the equivalence $\nabla_{x_t} \ln p_t(x_t) = -\epsilon_t/\sqrt{1 - \bar{\alpha}_t}$. Any model trained to predict the noise can be written in terms of the score, which is an essential property of our work. Whenever we derive some expression with respect to the score, we can use the noise-based formulation for forward and reverse diffusion processes by simply substituting $\epsilon_t = -\sqrt{1 - \bar{\alpha}_t}\nabla_{x_t} \ln p_t(x_t)$.

## B  Normalisation of Forces

Using the relationship between energy and force, we start with the equation:

$$m_i \frac{d^2 x_i}{dt^2} = F_i = -\frac{\partial}{\partial x_i} U(x_1, x_2, \ldots, x_N), \tag{20}$$

where $F_i$ is the force acting on the $i$-th atom, $m_i$ is the mass of the $i$-th atom, $x_i$ is the position vector of the $i$-th atom, and $U$ is the energy as a function of the positions of all $N$ atoms. Let $F = \{f_1, f_2, \ldots, f_N\}$ be a set of 3-dimensional vectors representing the forces acting on $N$ atoms. Each vector $f_i \in \mathbb{R}^3$ consists of the force components along the $x$, $y$, and $z$ coordinates for the $i$-th atom. We rescale each vector $f_i$ such that its magnitude does not exceed a predefined maximum norm while maintaining its direction. The process involves three main steps:

**Norm Calculation**   Compute the Euclidean norm (or $L^2$ norm) of each vector $f_i$. The Euclidean norm of $f_i$ is defined as:

$$\|f_i\|_2 = \sqrt{f_{i,x}^2 + f_{i,y}^2 + f_{i,z}^2}, \tag{21}$$

where $f_{i,x}$, $f_{i,y}$, and $f_{i,z}$ represent the components of the $i$-th vector $f_i$ along the $x$, $y$, and $z$ axes, respectively.

**Normalization**   Normalize each vector $f_i$ to obtain a unit vector $\hat{f}_i$ by dividing it by its norm. To avoid division by zero, a small constant $\epsilon = 1e-6$ is added to the norm. The normalization step is described as:

$$\hat{f}_i = \frac{f_i}{\|f_i\|_2 + \epsilon}. \tag{22}$$

**Rescaling**   Multiply each normalized vector $\hat{f}_i$ by a predefined maximum norm (we set the maximum norm to 1):

$$f_{i,\text{rescaled}} = \hat{f}_i \times \text{max\_norm}. \tag{23}$$

The output is a set of rescaled force vectors $F = \{f_{1,\text{rescaled}}, f_{2,\text{rescaled}}, \ldots, f_{N,\text{rescaled}}\}$, where each 3-dimensional vector $f_{i,\text{rescaled}}$ maintains its original direction and has its components within the range of $-1$ to $1$, ensuring stability in the sampling algorithm.

## C   PHYSICAL INTERPRETATION

In molecular dynamics (MD) simulations, the energy of molecular complexes is typically measured in kilocalories per mole (*kcal/mol*). The derivative of energy with respect to spatial position, i.e., the force, is thus expressed in *kcal/mol/Å*, where Å denotes angstroms ($10^{-10}$ meters). This conversion from energy to force is important as it indicates both the magnitude and direction of forces exerted on atoms, facilitating the prediction of atomic movements over time within the simulation environment.

The relationship between the force applied to an atom and the resulting displacement can be understood through the basic kinematic equation:

$$\Delta x = 0.5 \times \left(\frac{F}{m}\right) \times \Delta t^2, \tag{24}$$

where $F$ is the applied force, $m$ is the mass of the atom, and $\Delta t$ is the duration of the timestep. This equation emphasizes that the displacement ($\Delta x$) of an atom is proportional to the applied force and the square of the time interval, and inversely proportional to the atom's mass.

Essentially, this process is similar to a diffusion process on the coordinates, with a mini one-step MD relaxation at every step, where the time-step size is determined by $\lambda_{sc}$. The size of the timestep can be inferred from the hyperparameter $\lambda_{sc}$. Thus, to simplify simulation calculations, a scaling factor for force, denoted as $\lambda_{sc}$, is introduced, representing the term $\frac{0.5 \times \Delta t^2}{m}$ from Equation 24. Assuming the mass of a typical carbon-alpha ($C_\alpha$) atom remains constant and that normalized forces $F$ range between $-1$ and $1$, the displacement for each simulation timestep can be efficiently computed as:

$$\Delta x = \lambda_{sc} \times F, \tag{25}$$

where $\lambda_{sc}$ is the hyperparameter that scales the forces. This relation allows re-interpreting our reverse diffusion process as a combination of a reverse DDPM step on the coordinates, coupled with a one-step MD relaxation at every step, where the time-step size is determined by $\lambda_{sc}$. The size of the timestep can be inferred from the hyperparameter $\lambda_{sc}$.

# D STRUCTURE RECONSTRUCTION

To calculate the energy, it is essential to reconstruct the full antibody-antigen complex $C$ along with the generated CDR region $R$. This process involves first reconstructing the complete 3D structure of the atoms in the CDR, following the pipeline outlined in (Luo et al., 2022). The reconstruction begins by determining the coordinates of the N, C, O, and side-chain $C_\beta$ atoms, which are positioned relative to the $C_\alpha$ location and orientation of each amino acid (Engh & Huber, 2012). After these core atoms are reconstructed, the remaining side-chain atoms are built using the side-chain packing function in Rosetta (Alford et al., 2017). Once the CDR region is restored, the full antibody-antigen complex $C$ is reconstructed. With the complete structure (including the antibody with its 6 CDRs and framework, as well as the antigen), the energy of the complex can be calculated. This process is then iteratively performed for $\lambda_{st} \times 100$ timesteps, assuming diffusion occurs over $t = 100$ timesteps. The iteration begins when forces are first applied at $\lambda_{st}$ and continues through the sampling process until the final timestep $t = 0$.

# E ALGORITHMS

The following subsections describe two additional algorithms that were initially considered alongside our primary method. However, due to the more promising initial results of the main method, we discontinued further experimentation with these alternatives.

## E.1 ALGORITHM 2: SAMPLING WITH FORCE GRADIENTS OF $x^t$

The initial sampling procedure is detailed in Algorithm 2 below.

---
**Algorithm 2** DIFFFORCE-$C_\alpha$ Sampling with Force Guidance
---
1: $x^T \sim \mathcal{N}(0, I)$
2: **for** $t = T, \ldots, 1$ **do**
3:      $z \sim \mathcal{N}(0, I)$ if $t > 1$, else $z = 0$
4:      $x^{t-1} = \frac{1}{\sqrt{\alpha_t}} \left( x^t - \frac{1-\alpha_t}{\sqrt{1-\bar{\alpha}_t}} \epsilon_\theta(x^t, t) \right) + \sigma_t z - \lambda_{sc} \mathbb{1}_{t \geq \lambda_{st}} \nabla_{x^t} U\left(x^t, s^t, O^t; C\right)$
5:      $R^{t-1} = \left(x^{t-1}, s^{t-1}, O^{t-1}\right)$, sample $s^{t-1}, O^{t-1}$ following (Luo et al., 2022)
6: **end for**
7: **return** $x^0, s^0, O^0$
---

## E.2 ALGORITHM 3: SAMPLING WITH FORCE GRADIENTS OF $x^0$ VIA APPROXIMATION

Another sampling procedure that was initially considered is detailed in Algorithm 3.

---
**Algorithm 3** DIFFFORCE-$C_\alpha$ Sampling with Force Guidance
---
1: $x^T \sim \mathcal{N}(0, I)$
2: **for** $t = T, \ldots, 1$ **do**
3:      $z \sim \mathcal{N}(0, I)$ if $t > 1$, else $z = 0$
4:      estimate $\hat{x}^0(R^t)$ using Eq 8, $\hat{s}^0(R^t)$ using Eq 12 and $\hat{O}^0(R^t)$ using Eq 14
5:      $x^{t-1} = \frac{1}{\sqrt{\alpha_t}} \left( x^t - \frac{1-\alpha_t}{\sqrt{1-\bar{\alpha}_t}} \epsilon_\theta(x^t, t) \right) + \sigma_t z - \lambda_{sc} \mathbb{1}_{t \geq \lambda_{st}} \nabla_{x^0} U\left(\hat{x}^0, \hat{s}^0, \hat{O}^0; C\right)$
6:      $R^{t-1} = \left(x^{t-1}, s^{t-1}, O^{t-1}\right)$, sample $s^{t-1}, O^{t-1}$ following (Luo et al., 2022)
7: **end for**
8: **return** $x^0, s^0, O^0$
---

# F DETAILS OF BASELINES

**DiffAb** DIFFAB (Luo et al., 2022) models CDR sequences and structures using a diffusion model. This approach represents the first use of deep learning to integrate antigen 3D structures into antibody

sequence-structure design, thereby enhancing specificity and efficacy. Results for DIFFAB are obtained by our own experiments.

**RAbD** The RosettaAntibodyDesign (RABD) (Adolf-Bryfogle et al., 2018) is a computational tool for antibody design that utilizes Rosetta energy functions. It employs a Monte Carlo plus minimization (MCM) approach, wherein changes in antibody sequence and structure are randomly sampled and optimized through energy minimization to enhance target specificity (Adolf-Bryfogle et al., 2018). Results for RABD are taken from a recent study Luo et al. (2022).

## G ENERGY LANDSCAPE: ADDITIONAL DETAILS AND PLOTS

The selection of hyperparameters $\lambda_{sc} = 0.05$ and $\lambda_{st} = 0.3$ for our study was guided by ablation studies examining the influence of the force start and force scale parameters in the DIFFFORCE model. The results are detailed in Figure 6, where the displayed values represent the mean. For example, to calculate the IMP metric for $\lambda_{st} = 0.1$, we averaged the samples of three HCDR regions: H1, H2, and H3. For each region, we computed the mean derived from $n = 25$ samples for the following combinations: $\lambda_{sc} = 0.01, \lambda_{st} = 0.1$; $\lambda_{sc} = 0.05, \lambda_{st} = 0.1$; and $\lambda_{sc} = 0.1, \lambda_{st} = 0.1$, across 19 test complexes.

The choice of $\lambda_{st} = 0.3$ was determined by averaging the optimal performance metrics for IMP and AAR, which peaked at $\lambda_{st} = 0.5$, and for RMSD, which was lowest at $\lambda_{st} = 0.1$. Similarly, $\lambda_{sc} = 0.05$ was selected because it provided the best outcomes for IMP and AAR at $\lambda_{sc} = 0.1$, while maintaining a lower RMSD value at $\lambda_{sc} = 0.01$. This ensured that multiple key metrics were optimized simultaneously. We argue that activating the forces earlier during sampling would enhance AAR and IMP metrics but result in longer sample generation times.

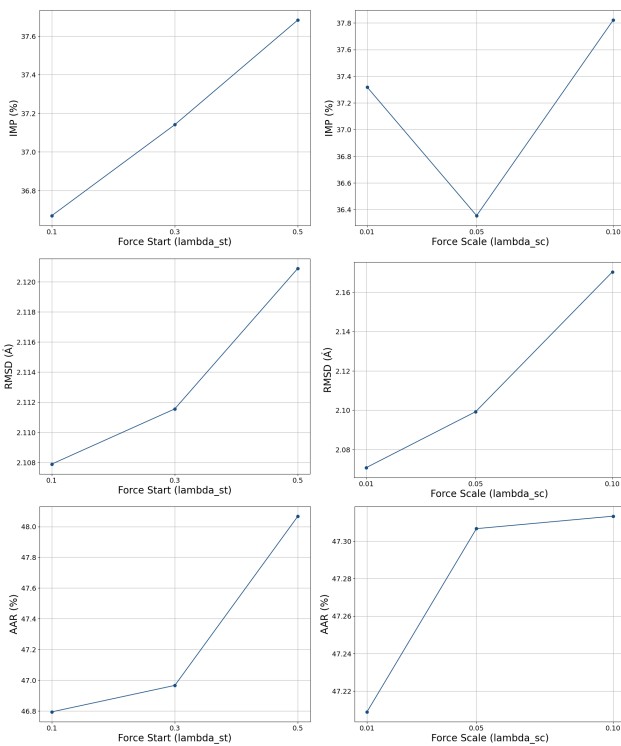

Figure 6: Ablation study showing the impact of the force start ($\lambda_{st}$) and force scale ($\lambda_{sc}$) hyperparameters on DIFFFORCE performance. The displayed values represent the mean. For IMP and AAR metrics, optimal results are obtained by activating forces early in the sampling process ($\lambda_{st} = 0.5$, 50% into sampling) with a higher force scale ($\lambda_{sc} = 0.1$). Conversely, for RMSD, better performance is achieved by activating forces later ($\lambda_{st} = 0.1$, 90% into sampling) with a lower force scale ($\lambda_{sc} = 0.01$).

Figure 7 provides an additional example from the experiment, analyzing three antigen-antibody complexes—PDB:7CHF, PDB:7CHE, and PDB:5TLK. The focus is on the heavy chain CDR regions, namely CDR-H1, CDR-H2, and CDR-H3. The figure demonstrates that the DIFFFORCE model, guided with force, generates antibody conformations with lower energy, indicating increased structural stability compared to DIFFAB.

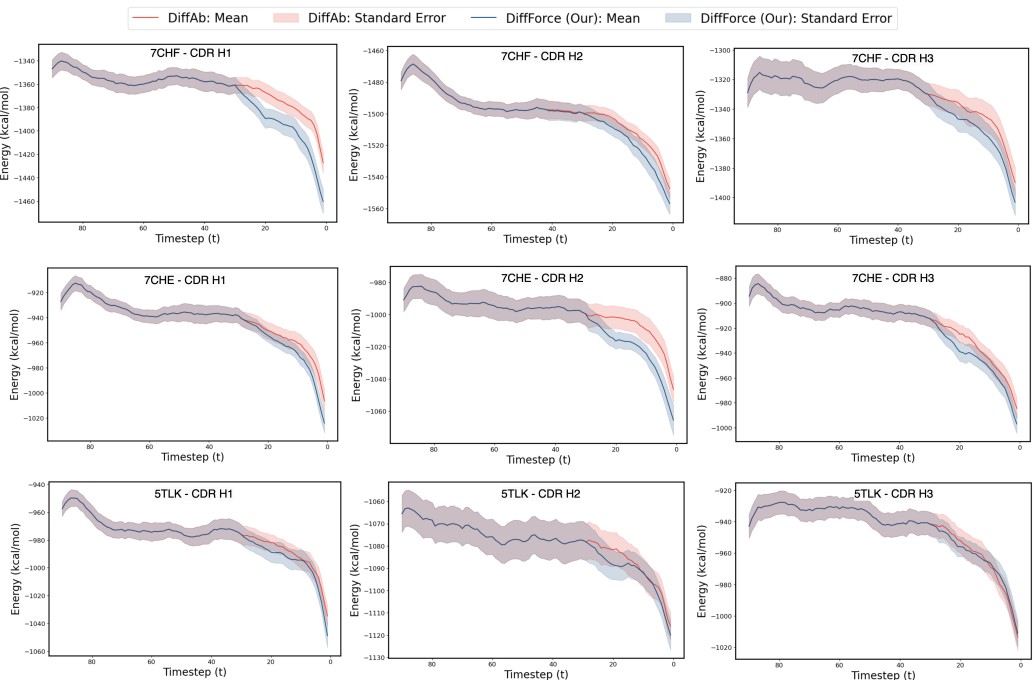

Figure 7: Energy landscape analysis of three antigen-antibody complexes—PDB:7CHF, PDB:7CHE, and PDB:5TLK—focused on the heavy chain CDR regions (CDR-H1, CDR-H2, and CDR-H3). The mean and standard error were derived from $n = 25$ samples across 25 seeds. The DIFFFORCE model, guided with force, converges to lower energy levels compared to DIFFAB.

## H   LIMITATIONS

**Computational Cost**   The iterative use of the MadraX library (Orlando et al., 2023) for force guidance during sampling is time-consuming due to the calculations involved. This process mimics molecular dynamics (MD) simulations to continuously update atomic positions based on various forces, including bond forces, electrostatic interactions, van der Waals forces, and solvent interactions. Each iteration estimates atomic movements, similar to gradient descent optimization steps. Thus, the computational demands should be considered when employing this approach.

**Reliability of Energy Function**   In our study, we utilized the Rosetta energy function (Alford et al., 2017) to evaluate the binding effectiveness of designed antibodies to their target antigens, a common metric in antibody design. Despite the integral role of Rosetta, along with tools such as FoldX (Schymkowitz et al., 2005), in simulating protein interactions, their reliability remains a subject of concern. These computational tools have been documented to exhibit inaccuracies when replicating experimental results, often due to the oversimplified models of complex molecular interactions they utilize (Ramírez & Caballero, 2016; 2018). This underscores the necessity for ongoing refinement of these computational methods.

**Evaluation Metrics**   In the field of antibody design, Amino Acid Recovery (AAR) and Root Mean Square Deviation (RMSD) are commonly used as evaluation metrics. However, these metrics have inherent limitations that may compromise the accurate assessment of an antibody's functional efficacy. AAR may not always reliably reflect the functional performance of the generated antibody sequences,

while RMSD primarily assesses the alignment of backbone atoms and overlooks the side chains, which are crucial for the specificity and strength of antigen-antibody interactions. These limitations underscore the need for the development of more comprehensive evaluation metrics.

## I BROADER IMPACTS

Integrating force guidance within the diffusion sampling process for antibody design can significantly accelerate therapeutic antibody discovery, with broad implications in fields like protein engineering. This advancement enables more precise modeling of atomic systems, enhancing predictions of protein stability, function, and interactions, crucial for designing enzymes and other biologically relevant molecules. However, potential societal drawbacks exist, particularly in dual-use applications. While aimed at therapeutic advancements, this approach could be misused to design harmful biological agents, raising ethical concerns and underscoring the need for regulations to ensure responsible use for societal benefit.

## J COMPUTE DETAILS

The sampling phase was performed using four NVIDIA A100-SXM-80GB GPUs. The relaxation stage was executed on an Intel(R) Xeon(R) Gold 6142 CPU @ 2.60GHz, equipped with 48 virtual cores and 256GB of RAM.

## K SOURCE CODE

The code will be made publicly available.

