# OpenReview forum: "Improving Antibody Design with Force-Guided Sampling in Diffusion Models"
_ICLR.cc/2025/Conference — ICLR 2025 Conference Withdrawn Submission_

### Official Review · Reviewer_ZBMN · 2024-11-02

**Soundness:** 2
**Presentation:** 3
**Contribution:** 1
**Rating:** 3
**Confidence:** 4

**Summary:**

The manuscript proposes a diffusion method with force-guidance for antibody design. Experimental results indicate that the proposed method can generate antibodies with low energy.

**Strengths:**

The paper is clear and easy to follow.

**Weaknesses:**

1. The method is a straightforward application of diffusion with guidance, and the theoretical derivations in the manuscript are simply replications of those from the original diffusion guidance method.

2. The method is compared with only a few baseline approaches on a limited set of tasks, despite many recent advancements in this area.

**Questions:**

What type of force field is used in the proposed method? How fast is the energy calculation?

---

### Official Review · Reviewer_LhWR · 2024-11-03

**Soundness:** 2
**Presentation:** 2
**Contribution:** 2
**Rating:** 3
**Confidence:** 4

**Summary:**

This paper introduces DiffForce, an extension of DiffAb (Luo et al. 2022), which adds a
physics-based force field to guide the reverse diffusion process for antigen-conditioned
antibody design. The integration of the force field builds on previous work by Komorowska
et al. (2024) but allows for residue-specific and orientation-specific terms due to a novel
method for predicting the final residue identities and orientations. The authors compare
the method to DiffAb and show that DiffForce results in improved amino acid recovery and
more energetically favourable complexes.
Currently, I would recommend to reject this paper for the following reasons: (1) It is
difficult to follow which parts of the theory are novel – most of the framework follows
DiffAb and equations 2-9 seem to come from Komorowska et al. (2024), despite claims like
“We have derived an approach...” (2) The predictions of the final residue identity and
orientation seem to be novel, but it is unclear how these are actually implemented and
there is a significant concern about data leakage (see below).

**Strengths:**

• The paper outlines a potentially promising approach for combining MD simulations
with pretrained diffusion models.
• The theory seems to be well-motivated, assuming we can estimate s0 and O0
effectively
• The improved energy scores show that the addition of MD is indeed resulting in
generated CDRs with lower energy.
• The sequence recovery is marginally better than that of DiffAb.

**Weaknesses:**

• While the energy demonstrations are important, they are not surprising considering
the only difference between this and DiffAb should be improved energy
minimization. It would also be useful to see plots in aggregate rather than individual
structures which can be cherry-picked.
• An important part of this work is the estimation of the final residues and
orientations, however the equations for these seem to depend on information which
should not be available at inference time. The incorporation of information from
earlier timesteps would give this model a significant unfair advantage. In particular:
o Eq. 12: Estimating sj0 conditioned on sj0?
o Eq. 14: Same question – we should not have access to Rs for s < t. Are we
performing a rollout of the reverse diffusion trajectory without the force field
at each time step?
• It is not clear that normalizing forces is appropriate as this may cause the structure
to exit stable equilibria where the magnitude of force should be small
• A motivation for incorporating physics is improved generalizability to unseen data,
however this is not tested in the experiments

**Questions:**

• (small) The use of computational methods is motivated by the ethical concerns of
animal testing – what about phage display?
• Eq. 2: “no explicit notion of the variable y” - isn’t “y” just “C” (conditioning on the
rest of the complex)
• 4.3.2: “particularly noticeable at earlier timesteps” - is this with T=100? What do
timesteps 20-30 look like?

---

### Official Review · Reviewer_UNs3 · 2024-11-03

**Soundness:** 3
**Presentation:** 3
**Contribution:** 2
**Rating:** 5
**Confidence:** 4

**Summary:**

This paper introduces a novel method to incorporate information from a physics-based force field to improve sampling quality of a structural antibody diffusion model.

**Strengths:**

The idea of incorporating information of a physics-based force field into ML-based sampling seems very promising, and could lead to substantial improvements in practical applicability of these methods which often generate clashes or physically impossible structures.

**Weaknesses:**

The benchmarking is relatively limited, comparing only to DiffAb, a very similar model without the guided sampling, and RAbD, a physics-based method. It would be useful to include comparisons with other recent ML work, such as dyMEAN, HERN, RFdiffusion or IgDiff. In table 1, it would be helpful to include standard deviation for each metric, to understand the statistical significance of these results, especially as numbers are given to 2 or 3 decimal points.
Though the incorporation of force field is very interesting, it is not completely novel and the experiments shown in this article are not completely convincing that this leads to significant and robust improvements in output quality.

**Questions:**

The authors focus on improvement to the binding affinity of generated antibodies, but the guiding force field is optimising for stability and overall energy rather than purely binding. Could the authors comment on how much the total energy of the protein/antibody improves when sampling with their guiding strategy, and whether this has other desirable properties, such as reducing the need for post-processing/relaxation?

In Algorithm 1, two hyperparameters are introduced that control the magnitude of the force applied during sampling. In the following section 4.2 and 4.3, a specific value is taken for both of these parameters, without a discussion of the tradeoff or impact of taking higher or lower coefficients. Could a discussion be added to expand on how table 1 would change for different parameter choices? There is a brief discussion in Appendix G, but it would be useful to provide a more qualitative understanding.

In Figure 4, it seems that the choice of when to apply the force field (at 70% in their experiments) plays an important role in changes to the output quality. Would starting the force field earlier in the sampling, or with a higher coefficient, substantially modify the results shown in this figure?

---

### Official Review · Reviewer_kGPP · 2024-11-04

**Soundness:** 2
**Presentation:** 2
**Contribution:** 3
**Rating:** 5
**Confidence:** 3

**Summary:**

This paper introduces DiffForce - approach and model allowing to sample from diffusion model with force field guidance. Authors benchmark their method in silico and demonstrate the improvements of sampled antibody sequences and structures across metrics like estimated binding free energy, native sequence and structure recovery.

**Strengths:**

This paper focuses on important subject - sampling sequence and corresponding structure of complementarity determining regions (CDRs) of antibodies in the context of binding partner (antigen). Improving this process can have important implications in drug discovery. The main original contribution of the paper is DiffForce - an model and algorithm which allows for guidance of the generation process with differentiable implementation of force field thus potentially allowing to skew the samples towards more physically plausible conformations.

**Weaknesses:**

While I believe that the idea and implementation described in the paper is valuable and has a significant potential I am not convinced by the evaluation. The paper has strong and general claims on "improving antibody design" and offering "enhanced quality of produced antibody sequences" but the results focus mostly on improvements of binding energies (estimated with the orthogonal, non-differentiable Rosetta force field). Sequence recovery and RMSD metrics are valuable metrics too but their interpretation is not as straightforward as higher / lower, respectively, equals better.

**Questions:**

In all, I believe that the manuscript is a strong starting point and an interesting contribution but authors should showcase more relevant benchmarks on the properties of generated sequences. Force fields have their own limitations, as authors correctly identify in the paper, and skewing the generation towards low energy samples can exploit their weaknesses and collapse into generating spurious examples of low quality. The results presented in Fig 3 and Fig 5 are to some extent anecdotal beyond improvement in estimated energies. With the low number of samples and even with the trained structural eye it is impossible to assess the extent of improvement.
The relevant literature contains multiple examples of benchmarking that could strengthen the conclusions shown in manuscript. In particular, I would welcome the orthogonal benchmarks including comparison of scores like structural fit analysis (e.g. through packing scores implemented in Rosetta), relevant properties of generated structures / sequences (like hydrophobicity, charge distributions which can assess whether model collapses to unrealistic, deteriorated samples), repeated patterns in sequences, likelihood of samples according to the LM etc.

---

### Author Response · Authors · 2024-11-21

Thank you for the feedback given. We are going to improve our paper according to your detailed suggestions.

---

### Note · Authors · 2024-11-21

I have read and agree with the venue's withdrawal policy on behalf of myself and my co-authors.